# Design, Synthesis and Biological Evaluation of Novel Osimertinib-Based HDAC and EGFR Dual Inhibitors

**DOI:** 10.3390/molecules24132407

**Published:** 2019-06-29

**Authors:** Hang Dong, Hao Yin, Chunlong Zhao, Jiangying Cao, Wenfang Xu, Yingjie Zhang

**Affiliations:** Department of Medicinal Chemistry, Key Laboratory of Chemical Biology (Ministry of Education), School of Pharmaceutical Sciences, Shandong University, 44 West Wenhua Road, Ji’nan 250012, China

**Keywords:** HDAC, EGFR, multi-target inhibitor, antitumor

## Abstract

Herein a novel series of histone deacetylases (HDACs) and epidermal growth factor receptor (EGFR) dual inhibitors were designed and synthesized based on the structure of the approved EGFR inhibitor osimertinib (AZD9291). Among them, four compounds **5D**, **5E**, **9D** and **9E** exhibited more potent total HDAC inhibition than the approved HDAC inhibitor SAHA. However, these compounds only showed moderate to low inhibitory potency towards EGFR with compounds **5E** and **9E** possessing IC_50_ values against EGFR^WT^ and EGFRT^790M^ in the micromolar range. 3-[4,5-dimethyl-2-thiazolyl]-2,5-diphenyl-2H-tetrazolium bromide (MTT) assay revealed the potent antiproliferative activities of compounds **5D**, **5E**, **9D** and **9E**, among which **9E** was even more potent against HeLa, MDA-MB-231, MDA-MB-468, HT-29 and KG-1 cell lines than SAHA and AZD9291. Further selectivity profile of **9E** showed that this compound was not active against other 13 cancer-related kinases and two epigenetic targets lysine specific demethylase 1 (LSD1) and bromodomain-containing protein 4 (BRD4). These results support further structural modification of **9E** to improve its EGFR inhibitory activity, which will lead to more potent and balanced HDAC and EGFR dual inhibitors as anticancer agents.

## 1. Introduction

The pathogenesis of tumors is extremely complex, and tumors have the characteristics of relapse and resistance [1,2]. With the intensive research on malignant tumors at the molecular level, various therapeutic targets have been discovered [3,4]. However, the clinical effectiveness of single-target drugs is usually transient, almost inevitably resulting in resistance and relapse due to the adaptability and heterogeneity of tumor cells and the tumor microenvironment [5,6]. Notably, multi-target drugs simultaneously targeting multiple biological molecules and multiple signal pathways involved in the development of tumors can produce additive or even synergistic antitumor effects, holding the promise of overcoming the problems of cancer resistance and relapse.

Epidermal growth factor receptor (EGFR) and its mediated signaling pathway regulate many physiological processes such as cell growth, proliferation and differentiation. Overexpression or mutation of EGFR plays an important role in the development, differentiation, survival and drug resistance of non-small-cell lung cancer (NSCLC) [7,8]. In recent years, EGFR has become a hot therapeutic target for cancer, especially for NSCLC [9]. Several reversible EGFR inhibitors including gefitinib (ZD1839) and erlotinib (CP358774) have been approved by the U.S. Food and Drug Administration (FDA) [10,11]. Another reversible EGFR inhibitor icotinib (BPI-2009) was approved by the China Food and Drug Administration (CFDA) in 2011 (Figure 1) [12]. These drugs have been widely used for advanced NSCLC patients, however, acquired resistance to these inhibitors frequently develops after a median of 9 to 13 months [13,14,15,16]. The common acquired resistance EGFR mutation with clinical implications is the T790M mutation, which dramatically decreases the binding affinity of reversible inhibitors in the EGFR active site. The EGFR T790M mutation is present in approximately 50% to 60% of resistant cases, the median survival is less than 2 years after the emergence of the T790M mutation [17,18]. To conquer the drug resistance resulting from the T790M mutation, much effort has been put on the development of irreversible EGFR inhibitors, which led to the approved drug osimertinib (Figure 1) [19]. However, resistance also arises rapidly, generally over a period of 9 to 13 months. The C797S mutation was reported to be a leading mechanism of resistance to the irreversible EGFR inhibitors targeting the T790M mutation [20].

It is worth noting that HDAC (histone deacetylase) inhibitors offer promise for overcoming the EGFR inhibitor resistance. For instance, it has been reported that HDAC inhibitors can decrease the expression of EGFR in colorectal cancer cells [21], and there are studies demonstrating that gefitinib combined with HDAC inhibitors synergistically induced growth inhibition and apoptosis in gefitinib-resistant NSCLC cells [22,23]. In addition, erlotinib combined with HDAC inhibitor showed a strong synergy in the suppression of cell growth by blocking the cell cycle and triggering cell apoptosis in EGFR-tyrosine kinase inhibitor (TKI)-resistant NSCLC cells, and this combined treatment led to a significant decrease in tumor growth and tumor weight compared with single agents alone [8]. The pan-HDAC inhibitor panobinostat sensitized lung adenocarcinoma cells to the antiproliferative effects of erlotinib [24]. Beyond that, some research showed that the HDAC inhibitor suberoylanilide hydroxamic acid (SAHA, Figure 2) remarkably inhibited cell viability and proliferation, induced cell apoptosis in EGFR-mutant lung cancer cell lines [25,26]. It was also reported that the combination of gefitinib and SAHA synergistically reduced cell growth and strongly induced apoptosis through inhibition of the insulin-like growth factor-1 receptor/protein kinase B (IGF-1R/AKT)-dependent signaling pathway [27]. These findings support that the combination of HDAC and EGFR inhibitors holds the promise of improving therapeutic effects and overcoming resistance in NSCLC patients. Currently, there is a phase I study of SAHA combined with gefitinib in patients with BIM polymorphysim associated resistant EGFR mutant lung cancer (ClinicalTrial.gov identifier: NCT02151721).

Compared with drug combination, multi-target drugs not only can attack multiple hallmark capabilities of cancer simultaneously but also can avoid the risks involved in drug combination, such as unpredictable pharmacokinetic profiles, drug-drug interactions, and poor patient compliance [28,29,30,31]. Recently, many HDAC inhibitor-based multi-target agents (Figure 2) have been developed by different research groups including ours, which showed promising in vitro and in vivo antitumor potency [32,33,34,35]. Moreover, the HDAC/EGFR/Her2 multi-target inhibitor CUDC-101 (Figure 2), which exhibited excellent in vitro and in vivo antitumor efficacies [36], has entered Phase I clinical trial.

Herein, on the basis of the prevalent design strategy of multi-target drugs and the promising anticancer potency of HDAC inhibitor and EGFR inhibitor combination, a novel series of osimertinib-based EGFR and HDAC dual inhibitors was rationally designed and readily synthesized. Among these analogs, compound **9E** was identified as a lead with high HDAC inhibitory activity, moderate EGFR inhibitory activity and desirable in vitro antiproliferative activity.

## 2. Results and Discussion

### 2.1. Compound Design

The proposed binding mode of osimertinib in EGFR revealed that the indole group and the 2-aminopyrimidine core could fit well into the ATP binding pocket, with the 2-aminopyrimidine core forming two important hydrogen bonds with Met793 in the hinge region. Therefore, modification of these two moieties could not be well tolerated for EGFR activity. The terminal aliphatic amine group is solvent-exposed, and the acrylamide is the functional group responsible for irreversible binding with Cys793 [37]. In our previous research, we successfully designed a series of HDAC and VEGFR dual inhibitors by introducing the zinc binding group (ZBG) of HDAC inhibitor pharmacophore to the solvent-exposed part of the VEGFR inhibitor pazopanib [32]. Here, the similar strategy as shown in Figure 3 was applied to design HDAC and EGFR dual inhibitors derived from EGFR inhibitor osimertinib. Note that the acrylamide group was deleted in the designed dual inhibitors based on the consideration that irreversible binding with EGFR will make the compound lose the opportunity to bind with HDAC.

### 2.2. Chemistry

The synthesis of compounds **5A**–**5E**, **10A** and **10B** is described in Scheme 1. The starting material **1** reacted with methyl 4-aminobenzoate hydrochloride to get **2**, which was hydrolyzed to obtain **3**. The carboxylic acid **3** was condensed with different methyl aminoalkanoates to obtain the key intermediates **4A**–**4E**. Compounds **4A**–**4E** and **2** then reacted with NH_2_OK in methanol to get the corresponding hydroxamic acid compounds **5A**–**5E** and **10A**, respectively. Intermediate **3** was condensed with o-phenylenediamine to afford the target compound **10B**.

Target compounds **9A**–**9E** were synthesized using the procedures described in Scheme 2. The starting material **1** reacted with 4-nitro-2-methoxyaniline under acidic conditions to give **6**. The intermediate **6** was reduced in the presence of iron powder, to obtain **7**. Intermediate **7** either reacted with cyclic anhydrides to give **8A** and **8B**, or was condensed with pimelic acid to give intermediate **8D**. The intermediates **8A**, **8B** and **8D** were converted to the target hydroxamic acid compounds **9A**, **9B**, **9D**, respectively. The intermediate **7** was also condensed with alkanedioic acid monoesters to afford intermediates **8C** and **8E**, which were converted to the target hydroxamic acid compounds **9C** and **9E**, respectively.

### 2.3. In Vitro HDAC and EGFR Inhibitory Assay

The HDAC inhibitory activities of all the target compounds were evaluated using HeLa cell nuclear extract (mainly contains HDAC1 and 2) as enzyme source. The results listed in Table 1 show that compounds **5C**–**5E**, **9C**–**9E** exhibited significant total HDAC inhibition with IC_50_ values lower than 1 μM. The exact IC_50_ values of the other compounds were not determined based on our in-house HDAC inhibitor standard that hydroxamic acid compound showing less than 50% inhibition at 1 μM and benzamide compound showing less than 50% inhibition at 10 μM are considered inactive. It was revealed that the linker length between ZBG and osimertinib scaffold had the most dramatic influence on HDAC inhibition. It was worth noting that compounds with linkers containing five or six carbons (**5D**, **5E**, **9D**, **9E**) were even more potent than the positive control SAHA. Considering osimertinib is a potent inhibitor against EGFR with T790M mutation, the EGFR inhibitory activities of all target compounds were preliminarily tested by determining the EGFR^T790M^ inhibitory rates at 1 μM. The results in Table 1 show that these osimertinib analogs possessed only moderate to low activities with inhibition rate ranging from 44% to 12%. The absence of acrylamide group for covalently binding to Cys797 in EGFR^T790M^ might account for their decreased activities relative to osimertinib.

Considering their compromised EGFR inhibitory potency, only two representative compounds **5E** and **9E** were further progressed to IC_50_ determination. The results in Table 2 show that these two compounds were active against both wild type EGFR and EGFR with T790M mutation. Compound **5E** was more potent than **9E**, which was in line with the results in Table 1.

### 2.4. In Vitro Antiproliferation Assay

Considering the potent HDAC inhibitory activities of **5D**, **5E**, **9D**, **9E**, these four compounds were tested in MTT assay to evaluate their antiproliferative activities (Table 3). Osimertinib and SAHA were used as the positive controls. It was demonstrated that our four compounds exhibited potent cytotoxicity to the several tested tumor cell lines. Notably, compared with SAHA and osimertinib, the most potent compound **9E** showed better antiproliferative activities against the human cervical carcinoma cell line HeLa, the human breast cancer cell lines MDA-MB-231, MDA-MB-468, the human colon carcinoma cell line HT-29 and the human acute myelogenous leukemia cell line KG-1, with IC_50_ values of 1.85, 0.60, 0.23, 0.79 and 0.24 μM, respectively.

### 2.5. Kinase Selectivity Profile

Compound **9E** was further tested against thirteen cancer-related kinases to characterize its protein kinase selective profile. The results in Figure 4 show that no significant inhibition with 100 nM of **9E** was observed.

### 2.6. LSD1 and BRD4 Assay

Lysine specific demethylase 1 (LSD1, also known as KDM1A) is a kind of epigenetic eraser, which catalyzes the oxidative demethylation of histone 3 methyllysine 4 (H3K4me1) and histone 3 dimethyllysine 4 (H3K4me2) [38]. It has been reported that LSD1 is overexpressed in various cancer cells and tissues, and regarded as a potential anticancer target [39]. A recent study revealed that AZD9291 could inhibit LSD1 with IC50 value of 3.98 ± 0.30 μM [40]. Therefore, compound **9E** was evaluated to see if this AZD9291 analog possessed promising LSD1 inhibitory activity. The result showed no significant inhibition at the concentration of 10 μM (Table 4).

There are research efforts indicating that some kinase inhibitors displayed strong inhibitory potential against BRD4, an epigenetic reader protein that recognizes acetylated histones and is also a potential anticancer target [41,42,43]. Based on this, compound **9E** was also tested in BRD4 binding assay, the result of which showed that **9E** was not active against BRD4 (BD1) at the concentration of 1 μM (Table 4).

## 3. Experimental Section

### 3.1. Chemistry

#### 3.1.1. Materials and Methods

The chemical reagents and solvents were purchased from commercial sources and were used without further purification. All reactions were monitored by thin-layer chromatography (TLC) on 0.25-mm silica gel plates (60GF-254). ^1^H NMR and ^13^C NMR spectra were obtained using a Bruker DRX spectrometer (Bruker, Billerica, MA, USA) at 600 and 150 MHz. ESI-MS data were recorded on an API 4000 spectrometer (SCIEX, Redwood City, CA, USA). High resolution mass spectra (HRMS) were conducted by Shandong Analysis and Test Center in Ji’nan. Melting points (Mp) were determined using open capillary on an uncorrected electrothermal melting point apparatus. Silica gel was used for column chromatography purification.

#### 3.1.2. Methyl 4-((4-(1-methyl-1H-indol-3-yl) pyrimidin-2-yl) amino) benzoate (**2**)

To a solution of methyl p-aminobenzoate hydrochloride (4.24 g, 22.60 mmol) and p-toluenesulfonic acid (3.90 g, 22.60 mmol) in 1, 4-dioxane (80 mL) was added **1** (5.0 g, 20.52 mmol). The reaction was monitored by TLC and stirred at 85 °C for 3 h. After cooling to room temperature, 6 mL of ammonia water was added dropwise followed by addition of 80 mL of H_2_O. The mixture was stirred at room temperature overnight with the resulting precipitate being filtered. The filter residue was washed with water and dried to give compound **2** (5.85 g, yield: 79%) as an orange-yellow solid, which was used in the following reaction without further purification. ^1^H NMR (600 MHz, DMSO-*d*_6_) δ 9.85 (s, 1H), 8.61 (d, *J* = 8.1 Hz, 1H), 8.42 (d, *J* = 5.3 Hz, 1H), 8.34 (s, 1H), 8.02 (d, *J* = 8.4 Hz, 2H), 7.93 (d, *J* = 8.4 Hz, 2H), 7.56 (d, *J* = 8.3 Hz, 1H), 7.33–7.28 (m, 2H), 7.24 (t, *J* = 7.5 Hz, 1H), 3.91 (s, 3H), 3.84 (s, 3H).

#### 3.1.3. 4-((4-(1-methyl-1H-indol-3-yl) pyrimidin-2-yl) amino) benzoic acid (**3**)

To a solution of compound **2** (2.0 g, 5.58 mmol) in CH_3_OH (40 mL) was added 4 M NaOH solution to adjust the pH to 14. The mixture was heated to reflux for 1 h. After cooling to room temperature, 1 M HCl solution was added to adjust the pH to neutral, the resulting precipitate was filtered, washed with water and dried to afford compound **3** (1.81 g, yield: 94%) as a pale yellow solid, which was used in the following reaction without further purification. ^1^H NMR (600 MHz, DMSO-*d*_6_) δ 12.42 (s,1H), 9.78 (s, 1H), 8.61 (d, *J* = 8.0 Hz, 1H), 8.41 (d, *J* = 5.4 Hz, 1H), 8.34 (s, 1H), 7.99 (d, *J* = 8.4 Hz, 2H), 7.90 (d, *J* = 8.4 Hz, 2H), 7.55 (m, 1H), 7.33–7.27 (m, 2H), 7.23 (t, *J* = 7.5 Hz, 1H), 3.91 (s, 3H).

#### 3.1.4. General Procedure for Preparation of **4A**–**4E**

*Methyl 3-(4-((4-(1-methyl-1H-indol-3-yl) pyrimidin-2-yl) amino) benzamido) propanoate* (**4A**). To a mixed solvent of DCM (40 mL) and DMF (4 mL) was added compound **3** (0.35 g, 1.0 mmol) followed by Et_3_N (0.20 g, 2.0 mmol) and TBTU (0.39 g, 1.2 mmol). The mixture was stirred in an ice bath for 0.5 h. Thereafter, methyl 3-aminopropionate hydrochloride (0.14 g, 1.2 mmol) was added, stirred at room temperature for 3 h. Then, the reaction solution was diluted with DCM (50 mL), washed with saturated NaHCO_3_ solution (20 mL), H_2_O (20 mL), and saturated NaCl solution (20 mL). The organic phase was dried over anhydrous MgSO_4_, filtered and condensed to afford crude product. The crude was triturated using petroleum ether, filtered and dried to afford a white powdery solid compound **4A** (0.35 g, yield: 80%). ^1^H NMR (600 MHz, DMSO-*d*_6_) δ 9.74 (s, 1H), 8.63 (d, *J* = 8.0 Hz, 1H), 8.44–8.38 (m, 2H), 8.36 (d, *J* = 2.8 Hz, 1H), 7.93 (d, *J* = 8.4 Hz, 2H), 7.81 (d, *J* = 8.4 Hz, 2H), 7.56 (d, *J* = 8.1 Hz, 1H), 7.32–7.27 (m, 2H), 7.24 (t, *J* = 7.6 Hz, 1H), 3.91 (s, 3H), 3.62 (s, 3H), 3.49 (m, 2H), 2.61 (t, *J* = 7.1 Hz, 2H).

Compounds **4B**–**4E** were prepared from compound **3** in a similar manner as described for compound **4A**.

*Methyl 4-(4-((4-(1-methyl-1H-indol-3-yl) pyrimidin-2-yl) amino) benzamido) butanoate* (**4B**). White powder solid compound **4B** (0.36 g, yield: 79%). ^1^H NMR (600 MHz, DMSO-*d*_6_) δ 9.65 (s, 1H), 8.62 (d, *J* = 8.0 Hz, 1H), 8.40 (d, *J* = 5.4 Hz, 1H), 8.32 (s, 1H), 8.27 (s, 1H), 7.93 (d, *J* = 8.4 Hz, 2H), 7.83 (d, *J* = 8.3 Hz, 2H), 7.55 (d, *J* = 8.2 Hz, 1H), 7.31–7.23 (m, 3H), 3.91 (s, 3H), 3.60 (s, 3H), 2.39 (t, *J* = 7.4 Hz, 2H), 1.81 (m, 2H), 1.25 (m, 2H).

*Methyl 5-(4-((4-(1-methyl-1H-indol-3-yl) pyrimidin-2-yl) amino) benzamido) pentanoate* (**4C**). White powder solid compound **4C** (0.28 g, yield: 57%). ^1^H NMR (600 MHz, DMSO-*d*_6_) δ 9.73 (s, 1H), 8.64 (d, *J* = 7.9 Hz, 1H), 8.40 (d, *J* =5.1 Hz, 1H), 8.35 (s, 1H), 8.34–8.31 (m, 1H), 7.96–7.92 (m, 2H), 7.85–7.81 (m, 2H), 7.56 (d, *J* = 8.1 Hz, 1H), 7.31–7.26 (m, 2H), 7.24 (t, *J* = 7.5 Hz, 1H), 3.90 (s, 3H), 3.59 (s, 3H), 2.69 (m, 2H), 2.36 (t, *J* = 7.1 Hz, 2H), 1.61–1.50 (m, 4H).

*Methyl 6-(4-((4-(1-methyl-1H-indol-3-yl) pyrimidin-2-yl) amino) benzamido) hexanoate* (**4D**). White powder solid compound **4D** (0.51 g, yield: 68%). ^1^H NMR (600 MHz, DMSO-*d*_6_) δ 9.72 (s, 1H), 8.64 (d, *J* = 8.1 Hz, 1H), 8.41–8.38 (m, 1H), 8.35 (d, *J* = 1.7 Hz, 1H), 8.32-8.28 (m, 1H), 7.93 (d, *J* = 8.3 Hz, 2H), 7.82 (d, *J* = 8.3 Hz, 2H), 7.56 (d, *J* = 8.2 Hz, 1H), 7.32–7.26 (m, 2H), 7.24 (t, *J* = 7.6 Hz, 1H), 3.90 (d, *J* = 1.7 Hz, 3H), 3.58 (d, *J* = 1.7 Hz, 3H), 3.27–3.22 (m, 2H), 2.32 (t, *J* = 7.5 Hz, 2H), 1.59–1.51 (m, 4H), 1.35–1.29 (m, 2H).

*Methyl 7-(4-((4-(1-methyl-1H-indol-3-yl) pyrimidin-2-yl) amino) benzamido) heptanoate* (**4E**). White powder solid compound **4E** (0.71 g, yield: 71%). ^1^H NMR(600 MHz, DMSO-*d*_6_) δ 9.62 (s, 1H), 8.62 (d, *J* = 6.6 Hz, 1H), 8.39 (d, *J* = 5.6Hz, 1H), 8.36 (s, 1H), 8.23–8.17 (m, 1H), 7.93 (d, *J* = 8.2 Hz, 2H), 7.83 (d, *J* = 7.9 Hz, 2H), 7.55 (d, *J* = 7.2 Hz, 1H), 7.32–7.22 (m, 3H), 3.91 (s, 3H), 3.59 (s, 3H), 3.27–3.24 (m, 2H), 2.33–2.27 (m, 2H), 1.58–1.50 (m, 4H), 1.35–1.30 (m, 4H). ESI-MS *m/z*: 486.34 [M + H]^+^.

#### 3.1.5. General Procedure for Preparation of **5A**–**5E**

*N-(3-(hydroxyamino)-3-oxopropyl)-4-((4-(1-methyl-1H-indol-3-yl) pyrimidin-2-yl) amino) benzamide* (**5A**). To a freshly prepared solution of potassium hydroxylamine (10 mL) compound **4A** (0.20 g, 0.47 mmol) was added, and the mixture was stirred at room temperature for 3 h. Then methanol was removed under reduced pressure, followed by addition of 10 mL of water. Then, 1 M hydrochloric acid was used to adjust the pH to 6, followed by filtration and water wash. The crude product was purified by flash column chromatography to afford pale yellow solid compound **5A** (0.11 g, yield: 55%). Mp: 164–167 °C. ^1^H NMR (600 MHz, DMSO-*d*_6_) δ 10.46 (s, 1H), 9.64 (s, 1H), 8.69 (s, 1H), 8.62 (d, *J* = 7.9 Hz, 1H), 8.40 (d, *J* = 5.2 Hz, 1H), 8.32 (s, 2H), 7.93 (d, *J* = 8.4 Hz, 2H), 7.83 (d, *J* = 8.4 Hz, 2H), 7.55 (d, *J* = 8.2 Hz, 1H), 7.27 (m, *J* = 23.8, 16.6, 7.4 Hz, 3H), 3.91 (s, 3H), 3.50–3.44 (m, 2H), 2.29 (m, *J* = 7.5 Hz, 2H). ^13^C NMR (150 MHz, DMSO-*d*_6_) δ 166.42, 162.68, 157.34, 144.14, 138.17, 138.17, 133.43, 128.28, 127.09, 126.01, 122.75, 121.38, 118.15, 110.88, 108.43, 36.49, 33.52, 33.11. HRMS (AP-ESI) *m/z* calcd for C_23_H_23_N_6_O_3_ [M + H]^+^ 431.1832, found 431.1827.

Compounds **5B**–**5E** were prepared from compounds **4B**–**4E** in a similar manner as described for compound **5A**.

*N-(4-(hydroxyamino)-4-oxobutyl)-4-((4-(1-methyl-1H-indol-3-yl) pyrimidin-2-yl) amino) benzamide* (**5B**). Yellow solid compound **5B** (0.12 g, yield: 47%). Mp: 165–168 °C. ^1^H NMR (600 MHz, DMSO-*d*_6_) δ 10.38 (s, 1H), 9.63 (s, 1H), 8.66 (s, 1H), 8.62 (d, *J* = 8.0 Hz, 1H), 8.40 (d, *J* = 5.3 Hz, 1H), 8.31 (s, 1H), 8.29 (d, *J* = 6.5 Hz, 1H), 7.94 (d, *J* = 8.4 Hz, 2H), 7.84 (d, *J* = 8.3 Hz, 2H), 7.55 (d, *J* = 8.1 Hz, 1H), 7.29 (t, *J* = 7.7 Hz, 1H), 7.28–7.21 (m, 2H), 3.90 (s, 3H), 2.05 (m, *J* = 7.6 Hz, 2H), 1.77 (m, *J* = 7.3 Hz, 2H). ^13^C NMR (150 MHz, DMSO-*d*_6_) δ 169.45, 166.43, 162.67, 160.26, 157.34, 144.06, 138.17, 133.41, 128.28, 127.27, 126.01, 122.75, 122.67, 121.38, 118.15, 112.96, 110.88, 108.41, 39.36, 33.52, 30.58, 25.93 (see the Appendix A). HRMS (AP-ESI) *m/z* calcd for C_24_H_25_N_6_O_3_ [M + H]^+^ 445.1988, found 445.1991.

*N-(5-(hydroxyamino)-5-oxopentyl)-4-((4-(1-methyl-1H-indol-3-yl) pyrimidin-2-yl) amino) benzamide* (**5C**). Yellow solid compound **5C** (0.11 g, yield: 55%). Mp: 144–147 °C. ^1^H NMR (600 MHz, DMSO-*d*_6_) δ 10.32 (s, 1H), 9.82 (s, 1H), 8.57 (d, *J* = 8.0 Hz, 1H), 8.38 (d, *J* = 6.5 Hz, 2H), 8.27 (d, *J* = 6.0 Hz, 1H), 7.90 (d, *J* = 8.5 Hz, 2H), 7.86 (d, *J* = 8.5 Hz, 2H), 7.56 (d, *J* = 8.2 Hz, 1H), 7.30 (d, *J* = 6.1 Hz, 2H), 7.24 (t, *J* = 7.6 Hz, 1H), 3.91 (s, 3H), 3.27 (m, *J* = 6.5 Hz, 2H), 2.00 (m, *J* = 7.5 Hz, 2H), 1.51–1.59 (m, 4H). ^13^C NMR (150 MHz, DMSO-*d*_6_) δ 169.53, 166.23,144.54,143.64,143.34, 138.27,134.30, 128.31, 125.99, 122.96, 122.69, 121.65, 118.95, 112.86, 111.01, 108.22, 39.35, 33.62, 32.55, 29.41, 23.26. HRMS (AP-ESI) *m/z* calcd for C_25_H_27_N_6_O_3_ [M + H]^+^ 459.2145, found 459.2153.

*N-(6-(hydroxyamino)-6-oxohexyl)-4-((4-(1-methyl-1H-indol-3-yl) pyrimidin-2-yl) amino) benzamide* (**5D**). Yellow solid compound **5D** (0.21 g, yield: 44%). Mp: 168–170 °C. ^1^H NMR (600 MHz, DMSO-*d*_6_) δ 10.34 (s, 1H), 9.93 (s, 1H), 8.57 (d, *J* = 8.1 Hz, 1H), 8.43 (s, 1H), 8.38 (d, *J* = 5.6 Hz, 1H), 8.34–8.28 (m, 1H), 7.87 (t, *J* = 7.3 Hz, 4H), 7.57 (d, *J* = 8.2 Hz, 1H), 7.33–7.29 (m, 2H), 7.23 (t, *J* = 7.6 Hz, 1H), 3.91 (s, 3H), 3.45 (m, *J* = 7.0 Hz, 2H), 1.97 (m, *J* = 7.5 Hz, 2H), 1.54 (m, *J* = 7.5 Hz, 4H), 1.31 (m, *J* = 7.9 Hz, 2H). ^13^C NMR (150 MHz, DMSO-*d*_6_) δ 169.55, 166.17, 163.67, 158.76, 143.07, 138.27, 134.73, 128.34, 128.27, 125.96, 123.01, 122.71, 121.74, 119.18, 112.77, 111.08, 108.13, 105.89, 104.50, 39.53, 33.67, 32.74, 29.51, 26.64, 25.42. HRMS (AP-ESI) *m/z* calcd for C_26_H_29_N_6_O_3_ [M + H]^+^ 473.2301, found 473.2299.

*N-(7-(hydroxyamino)-7-oxoheptyl)-4-((4-(1-methyl-1H-indol-3-yl) pyrimidin-2-yl) amino) benzamide* (**5E**). Yellow solid compound **5E** (0.16 g, yield: 53%). Mp: 161–163 °C. ^1^H NMR (600 MHz, DMSO-*d*_6_) δ 10.31 (s, 1H), 9.63 (s, 1H), 8.60 (d, *J* = 8.0 Hz, 1H), 8.39 (d, *J* = 5.4 Hz, 1H), 8.31 (s, 1H), 8.22 (d, *J* = 6.0 Hz, 1H), 7.92 (d, *J* = 8.4 Hz, 2H), 7.83 (d, *J* = 8.4 Hz, 2H), 7.55 (d, *J* = 8.2 Hz, 1H), 7.29 (t, *J* = 7.7 Hz, 1H), 7.27 (d, *J* = 5.5 Hz, 1H), 7.23 (t, *J* = 7.6 Hz, 1H), 3.90 (s, 3H), 3.26 (m, *J* = 6.9 Hz, 2H), 1.96 (m, *J* = 7.4 Hz, 2H), 1.52 (m, *J* = 7.7 Hz, 4H), 1.31 (m, *J* = 6.6 Hz, 4H). ^13^C NMR (150 MHz, DMSO-*d*_6_) δ 169.65, 166.33, 162.75, 160.11, 157.13, 143.91, 138.15, 133.53, 133.48, 128.26, 125.98, 122.79, 122.69, 121.42, 118.23, 112.88, 110.91, 108.36, 39.75, 33.53, 32.73, 29.64, 28.85, 26.72, 25.58. HRMS (AP-ESI) *m/z* calcd for C27H31N6O3 [M + H]^+^ 487.2458, found 487.2455.

Compound **10A** was prepared from compound **2** in a similar manner as described for compound **5A**.

*N-hydroxy-4-((4-(1-methyl-1H-indol-3-yl) pyrimidin-2-yl) amino) benzamide* (**10A**). Pale yellow solid compound **10A** (0.15 g, yield: 41%). Mp: 108–111 °C. ^1^H NMR (600 MHz, DMSO-*d*_6_) δ 11.13 (s, 1H), 9.74 (s, 1H), 8.63 (d, *J* = 8.1 Hz, 1H), 8.39 (d, *J* = 5.3 Hz, 1H), 8.34 (s, 1H), 7.94 (d, *J* = 8.4 Hz, 2H), 7.77 (d, *J* = 8.3 Hz, 2H), 7.54 (t, *J* = 9.0 Hz, 1H), 7.26 (m, *J* = 25.0, 7.6 Hz, 4H), 3.89 (s, 3H). ^13^C NMR (150 MHz, DMSO-*d*_6_) δ 162.68, 160.23, 157.32, 144.16, 138.16, 135.69, 133.45, 128.00, 126.00, 125.98, 125.29, 123.07, 122.75, 122.71, 121.84, 121.38, 118.50, 118.32, 118.27, 112.92, 112.85, 110.91, 110.88, 108.43. ^13^C NMR (150 MHz, DMSO-*d*_6_) δ 162.68, 160.23, 157.32, 144.16, 138.16, 133.45, 128.00, 126.00, 125.98, 123.07, 122.75, 121.38, 118.32, 118.27, 112.92, 112.85, 110.88, 108.43, 39.68. HRMS (AP-ESI) *m/z* calcd for C_20_H_18_N_5_O_2_ [M + H]^+^ 359.1508, found 359.1532.

#### 3.1.6. N-(2-aminophenyl)-4-((4-(1-methyl-1H-indol-3-yl) pyrimidin-2-yl)amino) benzamide (**10B**)

To a solution of compound **3** (0.35 g, 1.0 mmol) in DMF (4 mL) Et_3_N (0.20 g, 2.0 mmol) and TBTU (0.39 g, 1.2 mmol) were added at 0 °C. After stirred for 0.5 h, *o*-phenylenediamine (0.12 g, 1.1 mmol) was added. The reaction was stirred at 60 °C overnight. The reaction solution was cooled to room temperature with the solution being diluted by water and extract with ethyl acetate. The organic extract was washed with water and brine, dried over MgSO_4_, filtered and evaporated under vacuum. The crude product was purified by flash column chromatography to afford a pale yellow powdery solid compound **10B** (70 mg, yield: 16%). Mp: 164–168 °C. ^1^H NMR (600 MHz, DMSO-*d*_6_) δ 9.70 (s, 1H), 9.48 (s, 1H), 8.64 (d, *J* = 7.9 Hz, 1H), 8.42 (d, *J* = 5.3 Hz, 1H), 8.33 (s, 1H), 8.02–7.96 (m, 4H), 7.56 (d, *J* = 8.1 Hz, 1H), 7.32–7.24 (m, 3H), 7.21(d, *J* = 7.8 Hz, 1H), 6.98 (t, *J* = 7.7 Hz, 1H), 6.81 (d, *J* = 8.0 Hz, 1H), 6.63 (t, *J* = 7.6 Hz, 1H), 4.86 (s, 2H), 3.91 (s, 3H). ^13^C NMR (150 MHz, DMSO-*d*_6_) δ 165.40, 162.71, 160.23, 157.36, 144.45, 143.55, 138.17, 133.50, 128.98, 127.98, 127.04, 126.69, 126.01, 124.27, 122.76, 121.42, 118.08, 116.80, 116.66, 112.91, 110.92, 108.48, 33.55. HRMS (AP-ESI) *m/z* calcd for C_26_H_23_N_6_O [M + H]^+^ 435.1933, found 435.1942.

#### 3.1.7. N-(2-methoxy-4-nitrophenyl)-4-(1-methyl-1H-indol-3-yl) pyrimidin-2-amine (**6**)

To a solution of 2-methoxy-4-nitroaniline (8.28 g, 49.24 mmol) and *p*-toluenesulfonic acid (8.48 g, 49.24 mmol) in 1,4-dioxane (150 mL) was added **1** (10.0 g, 41.04 mmol) The reaction was stirred at 85 °C for 5 h. After cooling to room temperature, 10 mL of aqueous ammonia was added, followed by addition of 100 mL of H_2_O. The mixture was stirred at room temperature overnight with the resulting precipitate being filtered. The filter residue was washed with water and dried to afford an orange-yellow solid compound **6** (12.60 g, yield: 81%). ^1^HNMR (600 MHz, DMSO-*d*_6_) δ 8.76 (d, *J* = 9.0 Hz, 1H), 8.50–8.44 (m, 2H), 8.40 (d, *J* = 3.7 Hz, 2H), 8.00 (d, *J* = 9.1 Hz, 1H), 7.87–7.82 (m, 1H), 7.57 (d, *J* = 8.3 Hz, 1H), 7.42 (d, *J* = 5.4 Hz, 1H), 7.28 (dt, *J* = 29.9, 7.4 Hz, 2H), 4.07 (s, 3H), 3.92 (s, 3H).

#### 3.1.8. 2-methoxy-N^1^-(4-(1-methyl-1H-indol-3-yl) pyrimidin-2-yl) benzene-1,4-diamine (**7**)

To a mixed solvent of EtOH (240 mL) and H_2_O (80 mL), compound **6** (6.0 g, 16 mmol) was added, followed by addition of reduced iron powder (3.57 g, 64 mmol) and NH_4_Cl (3.43 g, 64 mmol). The mixture was heated to reflux and stirred for 3 h, then filtered with diatomaceous earth. The filtrate was evaporated under reduced pressure, then 200 mL of dichloromethane was added and dried over MgSO_4_. After filtration, the filtrate was condensed under reduced pressure to afford a brown oil compound **7** (5.02 g, yield: 90%). ^1^H NMR (600 MHz, DMSO-*d*_6_) δ 8.40 (d, *J* = 8.2 Hz, 1H), 8.22 (s, 1H), 8.18 (d, *J* = 5.3 Hz, 1H), 7.70 (s, 1H), 7.49 (d, *J* = 8.2 Hz, 1H), 7.45 (d, *J* = 8.3 Hz, 1H), 7.23 (t, *J* = 7.6 Hz, 1H), 7.11 (t, *J* = 7.5 Hz, 1H), 7.04 (d, *J* = 5.3 Hz, 1H), 6.36 (d, *J* = 2.2 Hz, 1H), 6.23–6.20 (m, 1H), 4.94 (s, 2H), 3.86 (s, 3H), 3.72 (s, 3H).

#### 3.1.9. General Procedure for Preparation of **8A** and **8B**

*4-((3-methoxy-4-((4-(1-methyl-1H-indol-3-yl) pyrimidin-2-yl) amino) phenyl) amino)-4-oxobutanoic acid* (**8A**). To a solution of succinic anhydride (0.20 g, 2 mmol) in dichloromethane (25 mL) was added compound **7** (0.69 g, 2 mmol), and the reaction was stirred at room temperature for 6 h. After the reaction was over, the solvent was removed under reduced pressure, with the crude product being purified by flash column to afford yellow powdery solid compound **8A** (0.60 g, yield: 67%). ^1^H NMR (600 MHz, DMSO-*d*_6_) δ 12.16 (s, 1H), 10.08 (s, 1H), 8.47–8.37 (m, 2H), 8.23 (s, 1H), 7.82 (s, 1H), 7.59–7.53 (m, 2H), 7.30–7.21 (m, 2H), 7.17 (d, *J* = 9.0 Hz, 2H), 7.09 (s, 1H), 3.90 (s, 3H), 3.81 (s, 3H), 2.61–2.57 (m, 2H), 2.57–2.53 (m, 2H).

Compound **8B** was prepared from compound **7** in a similar manner as described for compound **8A**.

*5-((3-methoxy-4-((4-(1-methyl-1H-indol-3-yl) pyrimidin-2-yl) amino) phenyl) amino)-5-oxopentanoic acid* (**8B**). Yellow powdery solid compound **8B** (0.57 g, yield: 68%). ^1^H NMR (600 MHz, DMSO-*d*_6_) δ 9.90 (s, 1H), 8.41 (d, *J* = 8.1 Hz, 1H), 8.31–8.26 (m, 2H), 7.98 (d, *J* = 8.6 Hz, 1H), 7.94 (s, 1H), 7.52 (d, *J* = 8.2 Hz, 1H), 7.49 (d, *J* = 2.2 Hz, 1H), 7.26 (t, *J* = 7.6 Hz, 1H), 7.18–7.13 (m, 3H), 3.88 (s, 3H), 3.82 (s, 3H), 2.37 (t, *J* = 7.4 Hz, 2H), 2.30 (t, *J* = 7.3 Hz, 2H), 1.87–1.81 (m, 2H).

#### 3.1.10. General Procedure for Preparation of **8C**–**8E**

*Methyl 6-((3-methoxy-4-((4-(1-methyl-1H-indol-3-yl) pyrimidin-2-yl)amino)phenyl) amino)-6-oxohexanoate* (**8C**). To a solution of compound **7** (0.76 g, 2.2 mmol) in DMF (8 mL), HOBT (0.35 g, 2.6 mmol), EDCI (0.50 g, 2.6 mmol), DIEA (0.43 g, 3.3 mmol) and monomethyl ester of adipic acid (0.35 g, 2.2 mmol) were added, and the mixture was stirred at 60 °C for 4 h. After the reaction was over, the solution was diluted by water and extracted with dichloromethane. The organic phase was washed with water and brine, then dried over anhydrous MgSO_4_. The solvent was evaporated under reduced pressure with the residues being purified by flash column chromatography to afford compound **8C** (0.37 g, yield: 34%). ^1^H NMR (600 MHz, DMSO-*d*_6_) δ 9.91 (s, 1H), 8.42 (d, *J* = 8.0 Hz, 1H), 8.30 (s, 1H), 8.28 (d, *J* = 5.4 Hz, 1H), 8.00–7.96 (m, 2H), 7.57–7.49 (m, 2H), 7.25 (t, *J* = 7.7 Hz, 1H), 7.19–7.12 (m, 3H), 3.88 (s, 3H), 3.82 (s, 3H), 3.60 (s, 3H). 2.52–2.49 (m, 2H), 2.39–2.35 (m, 2H), 1.55–1.46 (m, 4H).

*7-((3-Methoxy-4-((4-(1-methyl-1H-indol-3-yl) pyrimidin-2-yl) amino) phenyl) amino)-7-oxoheptanoic acid* (**8D**). To a solution of pimelic acid (1.9 g, 12 mmol) in dichloromethane (50 mL), Et_3_N (0.67 g, 6 mmol) and TBTU (0.69 g, 3 mmol) were added. The mixture was stirred in ice bath for 0.5 h, then compound **7** (1.0 g, 3 mmol) was added. The reaction mixture was stirred at room temperature overnight. The reaction solution was washed with 1 M NaOH solution (25 mL × 2), then the aqueous layer was adjusted to pH = 4 with 1 M hydrochloric acid. The resulting precipitate was filtered, washed with water and dried to afford yellow powdery solid compound **8D** (0.30 g, yield: 21%).

Compound **8E** was prepared from compound **7** in a similar manner as described for compound **8C**.

*Methyl 8-((3-methoxy-4-((4-(1-methyl-1H-indol-3-yl) pyrimidin-2-yl) amino) phenyl) amino)-8-oxooctanoate* (**8E**). Yellow solid compound **8E** (1.11 g, yield: 46%). ^1^H NMR (600 MHz, DMSO-*d*_6_) δ 9.84 (s, 1H), 8.41 (d, *J* = 8.1 Hz, 1H), 8.30–8.27 (m, 2H), 7.99 (d, *J* = 8.6 Hz, 1H), 7.92 (s, 1H), 7.52 (d, *J* = 8.2 Hz, 1H), 7.50–7.48 (m, 1H), 7.27–7.23 (m, 1H), 7.18–7.13 (m, 3H), 3.88 (s, 3H), 3.83 (s, 3H), 3.59 (s, 3H), 2.31 (t, *J* = 7.4 Hz, 4H), 1.64–1.59 (m, 2H), 1.58–1.52 (m, 2H), 1.35–1.30 (m, 4H).

#### 3.1.11. General Procedure for Preparation of **9A**–**9D**

*N*^1^*-hydroxy-N*^4^*-(3-methoxy-4-((4-(1-methyl-1H-indol-3-yl) pyrimidin-2-yl) amino) phenyl) succinimide* (**9A**). To a solution of compound **8A** (0.22 g, 0.5 mmol) in THF (20 mL) Et_3_N (0.10 g, 1.0 mmol) was added. Isobutyl chloroformate (82 mg, 0.6 mmol) dissolved in THF (1 mL) was added to the reaction mixture under ice bath, then the mixture was stirred for 1 h in ice bath. The mixture of hydroxylamine hydrochloride (70 mg, 1 mmol) and KOH (56 mg, 1 mmol) in methanol (5 mL) was stirred for 5 min then poured directly into the reaction mixture. The reaction continued for 3 h at room temperature, then the solvent was removed under reduced pressure followed by addition of 15 mL of water. Then, 1 M hydrochloric acid was used to adjust the pH to 6. The resulting precipitate was filtered, washed with water to get the crude product, which was purified by flash column chromatography to afford yellow powder solid compound **9A** (0.10 g, yield: 48%). Mp: 151–153 °C. ^1^H NMR (600 MHz, DMSO-*d*_6_) δ 10.46 (s, 1H), 9.97 (s, 1H), 8.75 (s, 1H), 8.42 (d, *J* = 8.1 Hz, 1H), 8.31–8.26 (m, 2H), 7.98 (d, *J* = 8.5 Hz, 1H), 7.95 (s, 1H), 7.53–7.50 (m, 2H), 7.26 (t, *J* = 7.6 Hz, 1H), 7.18–7.12 (m, 3H), 3.88 (s, 3H), 3.82 (s, 3H), 2.57–2.60 (m, 2H), 2.31 (t, *J* = 7.3 Hz, 2H). ^13^C NMR (150 MHz, DMSO-*d*_6_) δ 170.37, 168.88, 162.61, 160.80, 157.46, 150.30, 138.08, 135.48, 133.24, 126.02, 124.60, 122.66, 122.63, 122.27, 121.23, 112.96, 111.07, 110.80, 107.49, 103.23, 56.05, 39.71, 33.47, 27.99. HRMS (AP-ESI) *m/z* calcd for C_24_H_25_N_6_O_4_ [M + H]^+^ 461.1937, found 461.1938.

Compounds **9B** and **9D** were prepared from compounds **8B** and **8D** in a similar manner as described for compound **9A**.

*N*^1^*-hydroxy-N*^5^*-(3-methoxy-4-((4-(1-methyl-1H-indol-3-yl) pyrimidin-2-yl) amino) phenyl) glutaramide* (**9B**). Pale yellow powder solid compound **9B** (0.30 g, yield: 58%). Mp: 146–148 °C. ^1^H NMR (600 MHz, DMSO-*d*_6_) δ 10.42 (s, 1H), 9.90 (s, 1H), 8.72 (s, 1H), 8.42 (d, *J* = 8.0 Hz, 1H), 8.28 (d, *J* = 6.8 Hz, 2H), 7.99 (d, *J* = 8.6 Hz, 1H), 7.94 (s, 1H), 7.54–7.48 (m, 2H), 7.26 (t, *J* = 7.6 Hz, 1H), 7.13–7.18 (m, 3H), 3.88 (s, 3H), 3.83 (s, 3H), 2.34 (t, *J* = 7.5 Hz, 2H), 2.04 (t, *J* = 7.5 Hz, 2H), 1.84 (m, *J* = 7.5 Hz, 2H). ^13^C NMR (150 MHz, DMSO-*d*_6_) δ 170.99, 169.29, 162.68, 160.87, 157.45, 150.42, 138.06, 135.51, 133.25, 126.02, 122.63, 121.23, 112.97, 121.23, 112.97, 111.24, 110.80, 107.51, 103.40, 103.28, 56.08, 36.14, 33.47, 32.17, 21.69. HRMS (AP-ESI) *m/z* calcd for C_25_H_27_N_6_O_4_ [M + H]^+^ 475.2094, found 475.2141.

#### 3.1.12. General Procedure for Preparation of **9C** and **9E**

*N*^1^*-hydroxy-N*^6^*-(3-methoxy-4-((4-(1-methyl-1H-indol-3-yl) pyrimidin-2-yl) amino) phenyl) adipamide* (**9C**). To a solution of potassium hydroxyamine in methanol (15 mL), compound **8C** (0.35 g, 0.72 mmol) was added. The mixture was evaporated under reduced pressure, then 15 mL of H_2_O was added, the mixture was adjusted to pH 6 with 1 M hydrochloric acid, filtered, washed with water. The crude product was purified by flash column chromatography to afford a pale yellow powdery solid compound **9C** (0.11 g, yield: 31%). Mp: 118–120 °C. ^1^H NMR (600 MHz, DMSO-*d*_6_) δ 10.43 (s, 1H), 9.90 (s, 1H), 8.75 (s, 1H), 8.43 (d, *J* = 8.0 Hz, 1H), 8.30–8.24 (m, 2H), 8.01 (d, *J* = 8.5 Hz, 1H), 7.97 (s, 1H), 7.51 (d, *J* = 8.2 Hz, 2H), 7.26 (t, *J* = 7.6 Hz, 1H), 7.19 (d, *J* = 8.7, 2.2 Hz, 1H), 7.13–7.17 (m, *J* = 6.3, 2H), 3.86 (s, 3H), 3.83 (s, 3H), 2.34 (t, *J* = 6.9 Hz, 2H), 2.02 (t, *J* = 6.9 Hz, 2H), 1.66–1.54 (m, 4H). ^13^C NMR (150 MHz, DMSO-*d*_6_) δ 171.31, 169.52, 162.62, 160.82, 157.45, 150.37, 150.29, 138.09, 135.53, 133.21, 126.02, 124.69, 122.65, 122.23, 121.24, 112.99, 111.24, 110.78, 107.51, 103.27, 56.07, 39.54, 36.72, 33.44, 32.70, 25.41. HRMS (AP-ESI) *m/z* calcd for C_26_H_29_N_6_O_4_ [M + H]^+^ 489.2250, found 489.2242.

*N*^1^*-hydroxy-N*^7^*-(3-methoxy-4-((4-(1-methyl-1H-indol-3-yl) pyrimidin-2-yl) amino) phenyl) heptanediamide* (**9D**). Yellow solid compound **9D** (0.11 g, yield: 46%). Mp: 124–128 °C. ^1^H NMR (400 MHz, DMSO-*d*_6_) δ 10.39 (s, 1H), 10.03 (s, 1H), 8.66 (s, 2H), 8.44 (s, 1H), 8.39 (d, *J* = 7.9 Hz, 1H), 8.24 (d, *J* = 5.7 Hz, 1H), 7.83 (d, *J* = 8.6 Hz, 1H), 7.58–7.52 (m, 2H), 7.28 (t, *J* = 7.6 Hz, 1H), 7.19–7.25 (m, 2H), 7.15 (t, *J* = 7.5 Hz, 1H), 3.90 (s, 3H), 3.81 (s, 3H), 2.34 (m, *J* = 7.4 Hz, 2H), 1.98 (m, *J* = 7.3 Hz, 2H), 1.62 (m, *J* = 7.3 Hz, 2H), 1.53 (m, *J* = 7.5 Hz, 2H), 1.35–1.28 (m, 2H). ^13^C NMR (150 MHz, DMSO) δ 171.56, 169.52, 158.46, 138.23, 126.00, 124.48, 123.20, 123.01, 122.85, 121.76, 121.73, 112.65, 111.35, 111.22, 111.04, 107.06, 103.40, 56.01, 45.84, 36.78, 33.68, 28.75, 25.37, 8.90. HRMS (AP-ESI) *m/z* calcd for C_27_H_31_N_6_O_4_ [M + H]^+^ 503.2407, found 503.2401.

Compound **9E** was prepared from compound **8E** in a similar manner as described for compound **9C**.

*N*^1^*-hydroxy-N*^8^*-(3-methoxy-4-((4-(1-methyl-1H-indol-3-yl) pyrimidin-2-yl) amino) phenyl) octanediamide* (**9E**). Yellow powder solid compound **9E** (0.20 g, yield: 57%). ^1^H NMR (600 MHz, DMSO-*d*_6_) δ 10.36 (s, 1H), 9.89 (s, 1H), 8.70–8.65 (m, 1H), 8.41 (d, *J* = 8.0 Hz, 1H), 8.30 (s, 1H), 8.27 (d, *J* = 5.4 Hz, 1H), 8.01–7.93 (m, 2H), 7.55–7.47 (m, 2H), 7.28–7.22 (m, 1H), 7.19–7.11 (m, 3H), 3.88 (s, 3H), 3.82 (s, 3H), 2.37 (t, *J* = 7.4 Hz, 2H), 2.30 (t, *J* = 7.3 Hz, 2H), 2.26–2.23 (m, 2H), 1.86–1.80 (m, 2H), 1.27–1.20 (m, 2H). ^13^C NMR (150 MHz, DMSO-*d*_6_) δ 171.46, 169.61, 162.72, 160.68, 157.16, 150.38, 138.12, 138.09, 135.68, 133.33, 126.04, 124.54, 122.65, 122.32, 121.25, 112.99, 111.26, 110.80, 107.49, 103.46, 56.10, 46.10, 39.67, 36.89, 33.47, 32.76, 28.92, 25.55, 8.97. HRMS (AP-ESI) *m/z* calcd for C_28_H_33_N_6_O_4_ [M + H]^+^ 517.2563, found 517.2567.

### 3.2. Biological Materials and Methods

#### 3.2.1. In Vitro HDACs Inhibition Fluorescent Assay

In vitro HDACs inhibition assays were conducted as previously described [44]. In brief, 10 μL of HeLa cell nuclear extract was mixed with different concentrations of tested compound (50 μL). The mixture was incubated at 37 °C for 5 min, followed by adding 40 μL of fluorogenic substrate Boc-Lys (acetyl)-AMC. After incubation at 37 °C for 30 min, the mixture was quenched by addition of 100 μL of developer containing trypsin and trichostatin A (TSA). Over another incubation at 37 °C for 20 min, fluorescence intensity was measured using a microplate reader (Molecular Devices, Sunnyvale, CA, USA) at excitation and emission wavelengths of 390 and 460 nm, respectively. The inhibition ratios were calculated from the fluorescence intensity readings of tested wells relative to those of control wells, and the IC_50_ values were calculated using a regression analysis of the concentration/inhibition data.

#### 3.2.2. In Vitro EGFR Inhibition

The assay was performed using Kinase-Glo Plus luminescence kinase assay kit. All of the enzymatic reactions were conducted at 30 °C for 40 min. The 50 µL reaction mixture in assay buffer contained 10 µM ATP, EGFR kinases, and different concentrations of compounds. Kinase activity was measured by quantitating the amount of ATP remaining in solution following a kinase reaction. The luminescent signal from the assay is correlated with the amount of ATP present and is inversely correlated with the amount of kinase activity. The IC_50_ values were calculated using nonlinear regression with normalized dose−response fit using Prism GraphPad software (GraphPad Software, San Diego, CA, USA).

#### 3.2.3. In Vitro Anti-Proliferative Assay

All cell lines (A549, HeLa, MDA-MB-231, MDA-MB-468, HT-29, KG-1, PC-3) were maintained in RPMI1640 medium containing 10% FBS at 37 °C in a 5% CO_2_ humidified incubator. Cell proliferation assay was determined by the MTT (3-[4,5-dimethyl-2-thiazolyl]-2,5-diphenyl-2H-tetrazolium bromide) method. Briefly, cells were passaged the day before dosing into a 96-well plate, allowed to grow for 12 h, and then treated with different concentrations of compound for 72 h. A 0.5% MTT solution was added to each well. After incubation for another 4 h, formazan formed from MTT was extracted by adding 200 μL of DMSO. Absorbance was then determined using a microplate reader at 570 nm.

#### 3.2.4. Kinases Inhibition Assay

The kinase inhibition assays were performed by Eurofins Cerep Corporation in France. In brief, evaluation of the effects of compounds on the activity of the human kinases was quantified by measuring the phosphorylation of the corresponding substrate using human recombinant enzymes and the LANCE detection method.

#### 3.2.5. LSD1 Inhibition Assay

Amplex red coupled LSD1 inhibition assay was carried out following reported method [45]. In brief, the compounds were incubated with the recombinant LSD1 and H3K4me2. After that, the fluorescence was measured at excitation wavelength 530 nm and emission wavelength 590 nm in order to evaluate the inhibition rate of the candidate compound.

#### 3.2.6. BRD4 Inhibition Assay

The assay was performed by TR-FRET technology using the recombinant BRD4 (BD1) and its corresponding ligand (BET). The TR-FRET signal from the assay is correlated with the amount of ligand binding to the bromodomain. All of the binding reactions were conducted at room temperature. The 20 µL reaction mixture in assay buffer contained bromodomains, BET ligand and the indicated amount of inhibitor. For the negative control (blank), 5 µL of the assay buffer was added instead of the BET ligand. The reaction mixture was incubated for 120 min, then TR-FRET signal was measured using Tecan Infinite M1000 plate reader (Tecan, Männedorf, Switzerland).

## 4. Conclusions

In summary, a novel series of AZD9291-based HDAC and EGFR dual inhibitors was designed, synthesized and evaluated. Among them, compound **9E** exhibited more potent HDAC inhibition than SAHA while less potent EGFR inhibition than AZD9291. The absence of acrylamide group for covalent binding with EGFR might account for the decreased EGFR inhibition of **9E** relative to AZD9291. Notably, compared with SAHA and AZD9291, **9E** showed superior antiproliferative activity against several tumor cell lines, especial against human breast cancer cell lines MDA-MB-231, MDA-MB-468 and human acute myelogenous leukemia cell line KG-1. Although preliminary selectivity profiling showed no significant inhibition of **9E** towards 13 selected kinases and two epigenetic targets, we cannot exclude the possibility of other off-target effects of **9E** responsible to its superior antiproliferative activity. Mechanism study and structural modification of **9E** are currently underway in our lab.

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
