# Peer review of "Design, Synthesis and Biological Evaluation of Novel Osimertinib-Based HDAC and EGFR Dual Inhibitors"

_molecules, 2019, doi:10.3390/molecules24132407_

Round 1

Reviewer 1 Report

This paper describes the synthesis of new hydroxamic acid derivatives, designed as osimetinib analogues and their biological activity evaluation. This work is worth to be published upon minor revision, since the results are interesting and pave the way to prepare compounds with improved biological interest. The English language needs to be checked throughout the text. A few examples, along with some minor points are listed below. Page 4, line 101: instead of “intermediate”, better use “carboxylic acid” Page 5, line 117: acidic conditions Page 5, line 118: …reduced in the presence of iron powder, to obtain 7. Page 2, line 60: The acronym SAHA should be explained (superoylanilide hydroxamic acid) and its structure should be given in Fig. 2. Table 1: suggestion to use another symbol, in compounds 5F, 5G, in order to indicate the absence of substituent X. Page 8, line 176: There are research efforts, Page 10, line 248: NMR data for 4E are not provided. Page 16: line 520: References Care should be taken on the presentation of this section.  In many cases, the Journals abbreviations are not correct (see for example Refs 8, 10, 11, 20 etc) and page numbers are missing (e. g. in refs 14, 18, 21, 25, 26 etc).

Author Response

Dear Reviewer,

We gratefully thank you for your constructive comments and suggestions that are very helpful to improve the quality of our original manuscript. In our revised manuscript, we carefully modified our manuscript item by item according to your valuable suggestions as follows:

Point 1: Page 4, line 101: instead of “intermediate”, better use “carboxylic acid”

Response 1: The “intermediate” has been corrected to “carboxylic acid” in our revised manuscript.

Point 2: Page 5, line 117: acidic conditions

Response 2: The “acidic condition” has been corrected to “acidic conditions”.

Point 3: Page 5, line 118: …reduced in the presence of iron powder, to obtain 7.

Response 3: The “reduced to obtain 7 by iron powder” has been replaced with “reduced in the presence of iron powder, to obtain 7” in our revised manuscript.

Point 4: The acronym SAHA should be explained (superoylanilide
hydroxamic acid) and its structure should be given in Fig. 2.

Response 4: The acronym SAHA has been explained (superoylanilide

hydroxamic acid), and its structure has been given in Fig. 2.

Point 5: Table 1: suggestion to use another symbol, in compounds 5F, 5G, in order
to indicate the absence of substituent X

Response 5: The symbol of compounds 5F, 5G have replaced with 10A, 10B.

Point 6: Page 8, line 176: There are research efforts.

Response 6: The “There are research” has been corrected to “There are research efforts”.

Point 7: Page 10, line 248: NMR data for 4E are not provided.

Response 7: The NMR data for 4E has been provided in our revised manuscript.

Point 8: Page 16: line 520: References

Response 8: The “Reference” has been corrected to “References”.

Point 9: Care should be taken on the presentation of this section. In many cases,
the Journals abbreviations are not correct (see for example Refs 8, 10,
11, 20 etc) and page numbers are missing (e. g. in refs 14, 18, 21, 25, 26
etc).

Response 9: The Journals abbreviations and the missing page numbers have been corrected in our revised manuscript.

Reviewer 2 Report

The paper by Hang Dong et al. reports the synthesis of a series of new double inhibitors based an EGFR inhibitor and the structural features of a HDAC inhibitor. The synthetic methods are straightforward and the assessment of the biological activity based on known standard methods, so that the results appear reliable.

The compounds prepared are endowed with strong antiproliferative activity comparable or better than the reference compounds, so that the paper is of interest.

However, the new compounds are good inhibitors of HDAC but have scarce effects on EGFR. Apart from the open question- which is the real mechanism that can justify the strong activity?- are these compounds just new potent HDAC inhibitors? The Authors should discuss this point. These data throw some questions about the way the Authors planned their project. They state (line 93, page 3) that the acrylamide group of osertinib was deleted in order to avoid the irreversible binding with EGFR that could impede the interaction with HDAC.  Then, why the title “Osertinib-based” compounds?

My concern is that this choice compromised the possibility of maintaining any activity against EGFR. Did the Authors have any evidence that a compound without any substituent in that position could be still active? Why did not they try to start from a compound which is a reversible inhibitor? The Authors should address these questions.

The literature about multitargeting compounds is somewhat outdated. A couple of more recent reviews (Musso et al. Biochemical Pharmacology 96 (2015) 297-305; Fu et al. European Journal of Medicinal Chemistry 136 (2017) 195-211) could be quoted.

There is a mistake in numbering references from Ref. 29 .

Scheme 2: correct monomethyl actenedioate

Author Response

We gratefully thank you for your constructive comments and suggestions that are very helpful to improve the quality of our original manuscript. In our revised manuscript, we carefully modified our manuscript item by item according to your valuable suggestions as follows:

Point 1: However, the new compounds are good inhibitors of HDAC but have scarce effects on EGFR. Apart from the open question- which is the real mechanism that can justify the strong activity?- are these compounds just new potent HDAC inhibitors? The Authors should discuss this point. These data throw some questions about the way the Authors planned their project. They state (line 93, page 3) that the acrylamide group of osertinib was deleted in order to avoid the irreversible binding with EGFR that could impede the interaction with HDAC. Then, why the title “Osertinib-based” compounds?

Response 1: We quite agree with your opinion that these compounds might have some off-target effects contributing to their potent antiproliferative activity. In order to probe the possible off-target effects, one representative compound 9E was progressed to selectivity profiling. Although preliminary selectivity profiling showed no significant inhibition of 9E towards 13 cancer-related kinases and two epigenetic targets (LSD1 and BRD4), we can not exclude the possibility of other off-target effects of 9E responsible to its superior antiproliferative activity. Mechanism study and structural modification of 9E are currently underway in our lab. Such discussion has been added in the conclusions part of our manuscript.

Apart from the acrylamide group, the scaffold of our designed compounds, which interacts with the hinge region of ATP binding pocket in EGFR, was from the structure of Osertinib. Therefore, we would like to title these analogs as Osertinib-based compounds.

Point 2: My concern is that this choice compromised the possibility of maintaining any activity against EGFR. Did the Authors have any evidence that a compound without any substituent in that position could be still active? Why did not they try to start from a compound which is a reversible inhibitor? The Authors should address these questions.

Response 2: To dispel your concern, we would like to show one example compound 7 (J. Med. Chem. 2013, 56, 7025−7048), which has no substituent in the acrylamide position but possessed potent inhibitory activity against mutant EGFR (IC50 = 9 nM).

Based on the results in our present manuscript and your valuable suggestion, we will try to develop HDAC and EGFR dual inhibitors from a reversible inhibitor, like the above compound 7. Thank you for your constructive comments again. (see attachement)

Point 3: The literature about multitargeting compounds is somewhat outdated. A couple of more recent reviews (Musso et al. Biochemical Pharmacology 96 (2015) 297-305; Fu et al. European Journal of Medicinal Chemistry 136 (2017) 195-211) could be quoted.

Response 3: According to your advice, we have updated the references in our revised manuscript.

Point 4: There is a mistake in numbering references from Ref. 29.

Response 3: The numbering mistake has been corrected in our revised manuscript.

Comment 5: Scheme 2: correct monomethyl actenedioate

Response 5: The “monomethyl actenedioate” has been corrected to “monomethyl suberate” in our revised manuscript.

Reviewer 3 Report

Overall  a well rounded paper with many new molecules and quite meticulous biological evaluation. Although not all results concerning activity are great, I recommend the publication of this paper after some minor English editing (some well placed commas are need it). It is my belief that these data can help researchers to better direct their research endeavors in the future.

 Please consider revising the following English language mishaps:

Line 50 offers

Line 68 drug combination, multi-target drug

Line 71 kinds?

Author Response

Dear Reviewer,

Thank you very much for your positive comments about our manuscript. Based on your advice, we have corrected the grammatical mistakes in our manuscript. We also carefully modified our manuscript item by item according to your valuable suggestions as follows:

Point 1: Line 50 offers 

Response 2: The “offers” has been corrected to “offer”.

Point 2: Line 68 drug combination, multi-target drug 

Response 2: The “multi-target drug” has been corrected to “multi-target drugs”.

Point 3: Line 71 kinds?

Response 3: The “kinds” has been corrected to “many”.